# Cancer cell adaptation to hypoxia involves a HIF-GPRC5A-YAP axis

Alexander Greenhough[1,2,†,*] iD, Clare Bagley[1,†], Kate J Heesom[3], David B Gurevich[4], David Gay[5], Mark Bond[6], Tracey J Collard[1], Chris Paraskeva[1], Paul Martin[4,7,8] iD, Owen J Sansom[5,9] iD, Karim Malik[2,**] iD & Ann C Williams[1,***] iD

## Abstract

Hypoxia is a hallmark of solid tumours and a key physiological feature distinguishing cancer from normal tissue. However, a major challenge remains in identifying tractable molecular targets that hypoxic cancer cells depend on for survival. Here, we used SILAC-based proteomics to identify the orphan G protein-coupled receptor GPRC5A as a novel hypoxia-induced protein that functions to protect cancer cells from apoptosis during oxygen deprivation. Using genetic approaches *in vitro* and *in vivo*, we reveal HIFs as direct activators of *GPRC5A* transcription. Furthermore, we find that GPRC5A is upregulated in the colonic epithelium of patients with mesenteric ischaemia, and in colorectal cancers high *GPRC5A* correlates with hypoxia gene signatures and poor clinical outcomes. Mechanistically, we show that GPRC5A enables hypoxic cell survival by activating the Hippo pathway effector YAP and its anti-apoptotic target gene *BCL2L1*. Importantly, we show that the apoptosis induced by GPRC5A depletion in hypoxia can be rescued by constitutively active YAP. Our study identifies a novel HIF-GPRC5A-YAP axis as a critical mediator of the hypoxia-induced adaptive response and a potential target for cancer therapy.

**Keywords** cancer; GPRC5A; HIF; hypoxia; YAP
**Subject Categories** Cancer; Vascular Biology & Angiogenesis

## Introduction

Hypoxia (reduced tissue $O_2$ levels) features prominently in pathophysiologies associated with a perturbed blood supply and is an important feature of solid tumours (Harris, 2002). Due to its cancer-specific nature and key regulatory role in tumour growth, hypoxia has been proposed as one of the best validated cancer-selective targets not yet exploited in oncology (Wilson & Hay, 2011). Intratumoral hypoxia occurs as the pace of tumour growth outstrips $O_2$ availability and is exacerbated by the developing tumour vasculature, which is often poorly formed with aberrant blood flow (Ruoslahti, 2002; McIntyre & Harris, 2015). Tumour cells in these hypoxic regions switch on an adaptive transcriptional response mediated primarily by the hypoxia-inducible factors (HIFs) that help them survive and continue to grow (Bottaro & Liotta, 2003; Pouyssegur *et al*, 2006). However, as transcriptional regulators are considered difficult therapeutic targets, an attractive prospect would be to identify druggable mediators of hypoxic cancer cell survival (Wilson & Hay, 2011). In this study, we identify a new hypoxia-activated GPCR signalling axis that enables colorectal tumour cells to survive the microenvironmental stress of hypoxia. We show that GPRC5A (G Protein-coupled Receptor Class C, Group 5, Member A)—an orphan GPCR of poorly understood regulation and function—is a bona fide transcriptional target of HIFs both *in vitro* and *in vivo*. Importantly, we show that upregulation of GPRC5A during hypoxia protects colorectal tumour cells from apoptosis by activating the Hippo pathway effector YAP. Our findings uncover a previously unappreciated role for GPRC5A as a key regulator of the adaptive response to hypoxia. This highlights a HIF-GPRC5A-YAP axis as a cancer cell vulnerability and an opportunity to exploit tumour-associated hypoxia for therapy.

1   Cancer Research UK Colorectal Tumour Biology Group, School of Cellular & Molecular Medicine, Faculty of Life Sciences, University of Bristol, Bristol, UK
2   Cancer Epigenetics Laboratory, School of Cellular & Molecular Medicine, Faculty of Life Sciences, University of Bristol, Bristol, UK
3   Proteomics Facility, Faculty of Life Sciences, University of Bristol, Bristol, UK
4   School of Biochemistry, Faculty of Life Sciences, University of Bristol, Bristol, UK
5   Cancer Research UK Beatson Institute, Glasgow, UK
6   School of Clinical Sciences, University of Bristol, Bristol, UK
7   School of Physiology, Pharmacology and Neuroscience, Faculty of Life Sciences,  University of Bristol, Bristol, UK
8   School of Medicine, Cardiff University, Cardiff, UK
9   Institute of Cancer Sciences, University of Glasgow, Glasgow, UK
    *Corresponding author (Lead contact). Tel: +44 (0) 117 331 2043; E-mail: a.greenhough@bristol.ac.uk
    **Corresponding author. Tel: +44 (0) 117 331 2078; E-mail: k.t.a.malik@bristol.ac.uk
    ***Corresponding author. Tel: +44 (0) 117 331 2070; E-mail: ann.c.williams@bristol.ac.uk
    †These authors contributed equally to this work

# Results

### Hypoxia induces GPRC5A protein expression

Regions of hypoxia are frequently found in colorectal cancers (Yoshimura *et al*, 2004; Dekervel *et al*, 2014), but previous omics studies of hypoxia-mediated cellular responses have largely focused on breast cancer cells (Mole *et al*, 2009; Djidja *et al*, 2014; Semenza, 2017). To identify novel hypoxia-induced proteins in colorectal cancer cells, we performed SILAC-based proteomics in SW620 cells grown in normoxia (21% $O_2$) or hypoxia (1% $O_2$). As shown in Fig 1A (and Appendix Table S1), these experiments confirmed hypoxia-induced expression of well-established HIF-regulated proteins (e.g. carbonic anhydrase 9; CA9, ~ 2.6-fold) as well as proteins not previously reported to be hypoxia-regulated, including GPRC5A (~ 1.6-fold). In line with our proteomics data, Western blotting confirmed GPRC5A to be induced by hypoxia (Fig 1B), apparent as a series of bands [likely due to dimerisation and post-translational modifications (Zhou & Rigoutsos, 2014)] that we verified the identity of using GPRC5A siRNA (Fig 1C, note the non-specific ~ 60 kDa band henceforth marked with an asterisk). As reported for other cell types (Tao *et al*, 2007), immunofluorescence revealed that GPRC5A localised to the plasma membrane in SW620 cells (Fig 1D). We confirmed the generality of GPRC5A upregulation by hypoxia using panel of malignant and pre-malignant human colorectal cell lines (Fig 1E). Although detectable levels of GPRC5A protein were present in normoxia, GPRC5A was robustly upregulated by hypoxia in all colorectal tumour cell lines tested (Fig 1E). These data identify GPRC5A as a novel hypoxia-induced protein.

### GPRC5A is a direct transcriptional target of HIFs

To address whether hypoxia-induced GPRC5A protein expression was dependent on HIFs we depleted major HIF-α isoforms (HIF-1α and HIF-2α). Low basal HIF-1/2α expression was detectable in normoxia, but was strongly induced by hypoxia (Fig 1F). As previously reported (Raval *et al*, 2005), CA9 expression was preferentially reduced by depletion of HIF-1α (Fig 1F). However, both basal and hypoxia-induced GPRC5A protein expression levels were diminished by individual HIF-1α or HIF-2α depletion (Figs 1F, and EV1A and B) and abolished by depletion of HIF-1/2α together (Fig 1F). This indicates that GPRC5A is a shared target of both HIF-1/2. Depletion of HIF-1β also diminished hypoxia-induced GPRC5A protein levels (Fig 1G), and induction of GPRC5A protein expression by hypoxia mimetic DMOG was HIF-1/2-dependent (Fig 1H). Quantitative transcript analysis following HIF-1/2α depletion confirmed that *GPRC5A* mRNA was also dependent on HIFs (Figs 1I and J, and EV1C and D). To assess whether *GPRC5A* represents a direct transcriptional target of HIFs, we performed ChIP-qPCR to ascertain HIF binding at the *GPRC5A* promoter using primers spanning an optimal (Wenger *et al*, 2005) hypoxia response element (HRE; 5′-B(A/G) CGTGVBBB-3′ [B = all bases except A; V = all bases except T]). In line with GPRC5A representing a HIF target, we found that HIF-1α bound a region of the *GPRC5A* promoter (−103/+47 relative to the TSS; Fig 1K) containing an optimal HRE (5′-CACGTGGCTT-3′, −58/−49), and binding of both HIF-1α and RNAPII to this region increased in hypoxia (Fig 1K). As controls, neither HIF-1α nor RNAPII were recruited to a downstream non-regulatory region of

the *GPRC5A* gene locus (+30,762/30,911) during hypoxia (Fig 1K). As positive controls, we confirmed hypoxia increased the binding of RNAPII and HIF-1α to the CA9 promoter (Fig 1K), as described previously. Taken together, these data strongly suggest that *GPRC5A* is a novel and direct HIF transcriptional target.

### GPRC5A is hypoxia/HIF-induced *in vivo*

Having identified GPRC5A as hypoxia/HIF-induced *in vitro*, it was important to investigate whether this is also true in an *in vivo* context. Firstly, we examined GPRC5A (and CA9) expression in human colorectal tissue samples from patients with mesenteric ischaemia, which is characterised by regions of acute $O_2$ deprivation (Kaidi *et al*, 2007). We validated antibodies for IHC using formalin-fixed paraffin-embedded hypoxic SW620 cells depleted of GPRC5A or CA9 (Fig 2A and B). In patients with mesenteric ischaemia, strong GPRC5A staining was present in the colonic epithelium, but not in normal tissue (Figs 2C and EV2). Using serial sections, CA9 staining mirrored GPRC5A expression, confirming hypoxia (Figs 2C and EV2). These findings support our *in vitro* data and suggest GPRC5A is induced by hypoxia *in vivo*. Secondly, to further establish the association of GPRC5A with HIFs and hypoxia *in vivo*, we took advantage of a mouse model where genes can be inducibly deleted specifically in intestinal epithelial cells (Sansom *et al*, 2004; Jackstadt & Sansom, 2016). It was previously noted that conditional Apc deletion in the intestine leads to Hif1a activation and increased carbonic anhydrase 9 expression (Newton *et al*, 2010). Quantitative transcript analyses of Apc-deleted (Villin-CreERT2 $Apc^{fl/fl}$) and Apc/Hif1a-deleted (Villin-CreERT2 $Apc^{fl/fl};Hif1a^{fl/fl}$) intestinal tissue revealed that *Hif1a* mRNA expression was reduced by ~ 70% following *Hif1a* deletion (Fig 2D). As expected, Hif1a preferred targets *Car9* and *Egln3* expression were reduced by *Hif1a* deletion (Fig 2D), but *Dll4* (a Hif2a target) was not affected (Fig 2D). These data indicate that *Gprc5a* is an *in vivo* physiological target of Hif1a in mouse intestinal epithelial cells. Interestingly, using an *in vivo* zebrafish model, we found that a related homologue (gprc5ba) was induced in a model of constitutive HIF activation (Fig 2E; Santhakumar *et al*, 2012) (Tg[*fli1*:eGFP;$vhl^{-/-}$]) and upon exposure of Tg[*fli1*:eGFP] zebrafish to hypoxia (Fig 2E and F). Finally, bioinformatic analysis on a transcriptomics dataset (GSE24551) from 320 primary colorectal cancers revealed that *GPRC5A* mRNA levels strongly correlated with HIF and hypoxia gene signatures (Fig 2G and H). Furthermore, we found that high *GPRC5A* transcripts closely correlated with poor survival outcomes in colorectal cancer patients (Fig 2I). However, while these data show an *in vivo* association between *GPRC5A*, hypoxia gene signatures and patient outcomes, it is important to note that this may be a reflection of *GPRC5A*'s regulation by HIF activity/ hypoxia in aggressive tumours, rather than necessarily indicating a functional role (Kaelin, 2017). Taken together, our *in vitro* and *in vivo* findings show that hypoxia and HIFs regulate *GPRC5A* and that high *GPRC5A* expression is an indicator of poor prognosis in colorectal cancer patients.

### GPRC5A promotes hypoxic cancer cell survival

The elevation of GPRC5A levels during hypoxia implicates it in adaptive signalling such as evasion of apoptosis. Exposure of colorectal cancer cells to hypoxia resulted in only a minor increase in apoptosis,

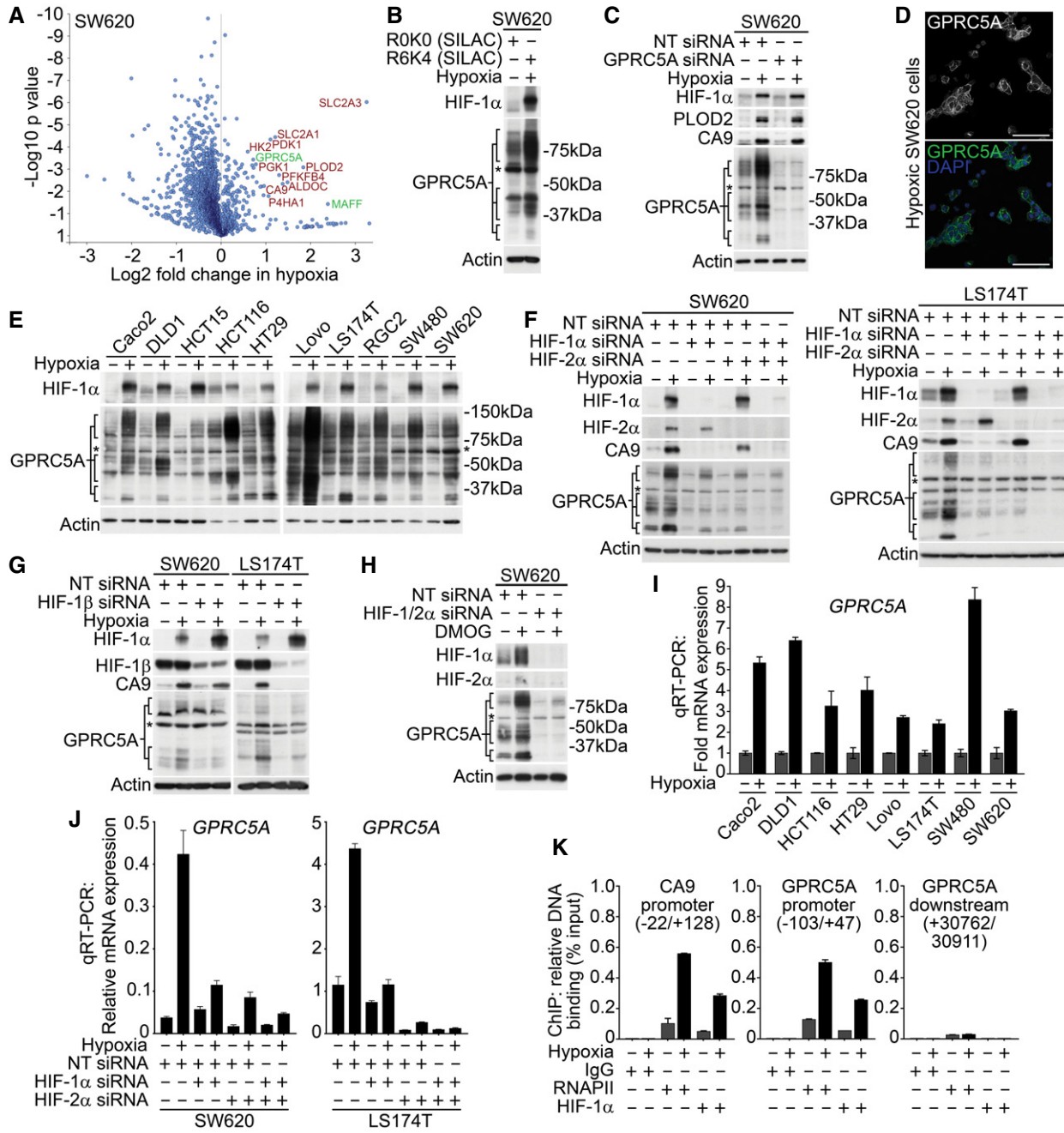

**Figure 1. Hypoxia induces GPRC5A directly via HIFs.**

A   SILAC-based proteomics data identify known (red) and novel (green) hypoxia-induced proteins in SW620 cells. One-sample *t*-test was performed.

B   Western blotting confirmed GPRC5A as a hypoxia-induced protein in SILAC lysates.

C   Validation of GPRC5A Western blot data using siRNA. *Non-specific band of ~60 kDa not depleted by GPRC5A siRNA.

D   Confocal microscopy showing plasma membrane GPRC5A expression in hypoxic SW620 cells (scale bars: 75 μm).

E   Western blotting showing GPRC5A upregulation by hypoxia in a panel of colorectal tumour cell lines.

F   Basal & hypoxia-induced GPRC5A protein expression was decreased by HIF-1/2α depletion.

G   Depletion of HIF-1β decreased GPRC5A protein upregulation in hypoxia.

H   Hypoxia mimetic DMOG induced HIF-1/2α, CA9 and GPRC5A protein expression. Dual HIF-1/2α depletion reduced GPRC5A induction by DMOG.

I   qRT–PCR demonstrating that *GPRC5A* mRNA was upregulated by hypoxia (n = 3). *GPRC5A* was normalised to *HPRT* (error bars ± SD).

J   qRT–PCR demonstrating that HIF-1/2α depletion decreased *GPRC5A* induction during hypoxia (n = 3). *GPRC5A* was normalised to *HPRT* (error bars ± SD).

K   ChIP–PCR analyses identify HIF-1α binding to the *GPRC5A* promoter region containing a putative optimal HRE (error bars ± SD, n = 3).

Data information: Asterisks (*) indicate non-specific band. Level adjustments were made to images in Adobe Photoshop post-acquisition for clarity (equal changes applied to the entire image). Representative examples of n = 3 independent experiments are shown.

Source data are available online for this figure.

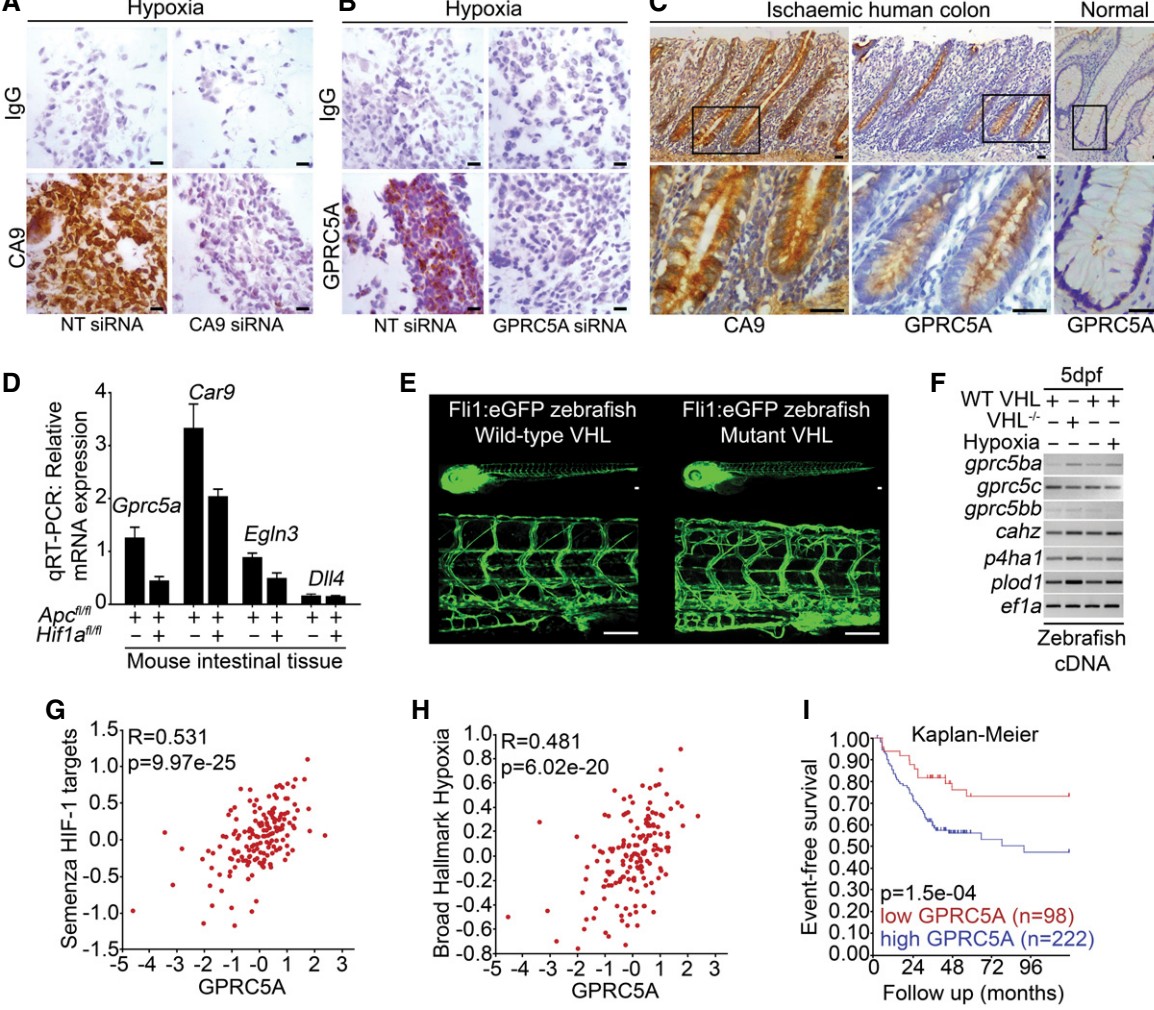

**Figure 2. GPRC5A is hypoxia/HIF-induced *in vivo*.**

A, B  Expression of CA9 and GPRC5A in formalin-fixed paraffin-embedded hypoxic SW620 cells by IHC. Reduced CA9 and GPRC5A expressions with siRNA confirm antibody specificity (scale bars: 200 μm).

C  IHC analysis of serial sections from human colorectal tissue from patients with mesenteric ischaemia (strangulated colon). GPRC5A is co-expressed with CA9 in the colonic epithelial cells (scale bars: 50 μm).

D  Quantitative RT–PCR analysis of mouse intestinal tissue. Gene expression was normalised to housekeeping gene *Tbp*. Raw data from three independent experiments (*n* = 3 mice) are shown (error bars ± SEM).

E  Tg[*fli1*:eGFP; *vhl*$^{−/−}$] and Tg[*fli1*:eGFP] zebrafish embryos (5 days post-fertilisation) demonstrate excessive angiogenesis and increased expression of HIF target genes (scale bars: 100 μm).

F  *gprc5ba* was induced in *vhl* mutant zebrafish embryos and *fli1*:eGFP zebrafish embryos exposed to 5% O$_2$ (vs. normoxia) for 24 h (RT–PCR).

G, H  Bioinformatic analysis of transcriptomics dataset GSE24551. Gene set analyses reveal *GPRC5A* mRNA strongly correlated with HIF/hypoxia gene signatures. GSEA datasets used were Semenza_HIF1_Targets (M12299) Broad_Hallmark_Hypoxia (M5891). Analysis was performed using R2 (http://r2.amc.nl).

I  Kaplan–Meier curve following analysis of transcriptomics dataset GSE24551. Event-free survival is significantly reduced in patients with tumours expressing high levels of *GPRC5A* mRNA. Analysis was performed using R2 (http://r2.amc.nl).

Data information: Level adjustments were made to images in Adobe Photoshop post-acquisition for clarity (equal changes applied to the entire image). Representative examples of *n* = 3 independent experiments are shown.

Source data are available online for this figure.

as determined by Western blotting for cleaved caspase-3 and PARP (Fig 3A). Strikingly, however, cells with depleted GPRC5A displayed substantial increases in these apoptotic markers upon exposure to hypoxia (Fig 3A). We confirmed these phenotypes with independent siRNAs that each caused similar hypoxia-specific increases in caspase-3/PARP cleavage (Fig 3B) and produced corresponding

reductions in hypoxic cell growth and survival (Fig 3C). To rule out off-target effects, we generated SW620 cells stably carrying a doxy-cycline-inducible siRNA-resistant (via synonymous mutations) GPRC5A (SW620:GPRC5A$^{si1R}$; Figs 3D and EV3A–C). Expression of GPRC5A$^{si1R}$ was resistant to siRNA-mediated knockdown (Fig 3D) and rescued increases in cleaved caspase-3 and PARP induced by

**Figure 3.  GPRC5A promotes hypoxic cancer cell survival.**

A   GPRC5A depletion markedly increases caspase-3 activation/PARP cleavage during hypoxia.

B   Three independent siRNA sequences targeting GPRC5A induce caspase-3 activation/PARP cleavage during hypoxia.

C   GPRC5A depletion reduces hypoxic cell growth/survival. Crystal violet cell assays show reduced cell growth/survival in GPRC5A-depleted cells during hypoxia ($n = 3$).

D   Expression of an siRNA-resistant GPRC5A cDNA rescues hypoxic GPRC5A-depleted cells from apoptosis. Upper: doxycycline-induced expression of GPRC5A$^{si1R}$ rescues increased caspase-3/PARP cleavage induced by GPRC5A depletion in hypoxia. Lower: generation of an siRNA1-resistant GPRC5A cDNA by synonymous mutations.

E   GPRC5A depletion in hypoxia induces apoptosis as determined by the violet ratiometric membrane asymmetry probe/dead cell apoptosis assay and flow cytometry ($n = 3$ independent experiments).

F   Caspase inhibitor QVD prevented caspase-3 activation/PARP cleavage by GPRC5A depletion in hypoxia.

Data information: Asterisks (*) indicate non-specific band. Level adjustments were made to images in Adobe Photoshop post-acquisition for clarity (equal changes applied to the entire image). Representative examples of $n = 3$ independent experiments are shown; data are presented as mean $\pm$ SEM. One-way ANOVA with Tukey's multiple comparisons test was carried out in (C and F).

Source data are available online for this figure.

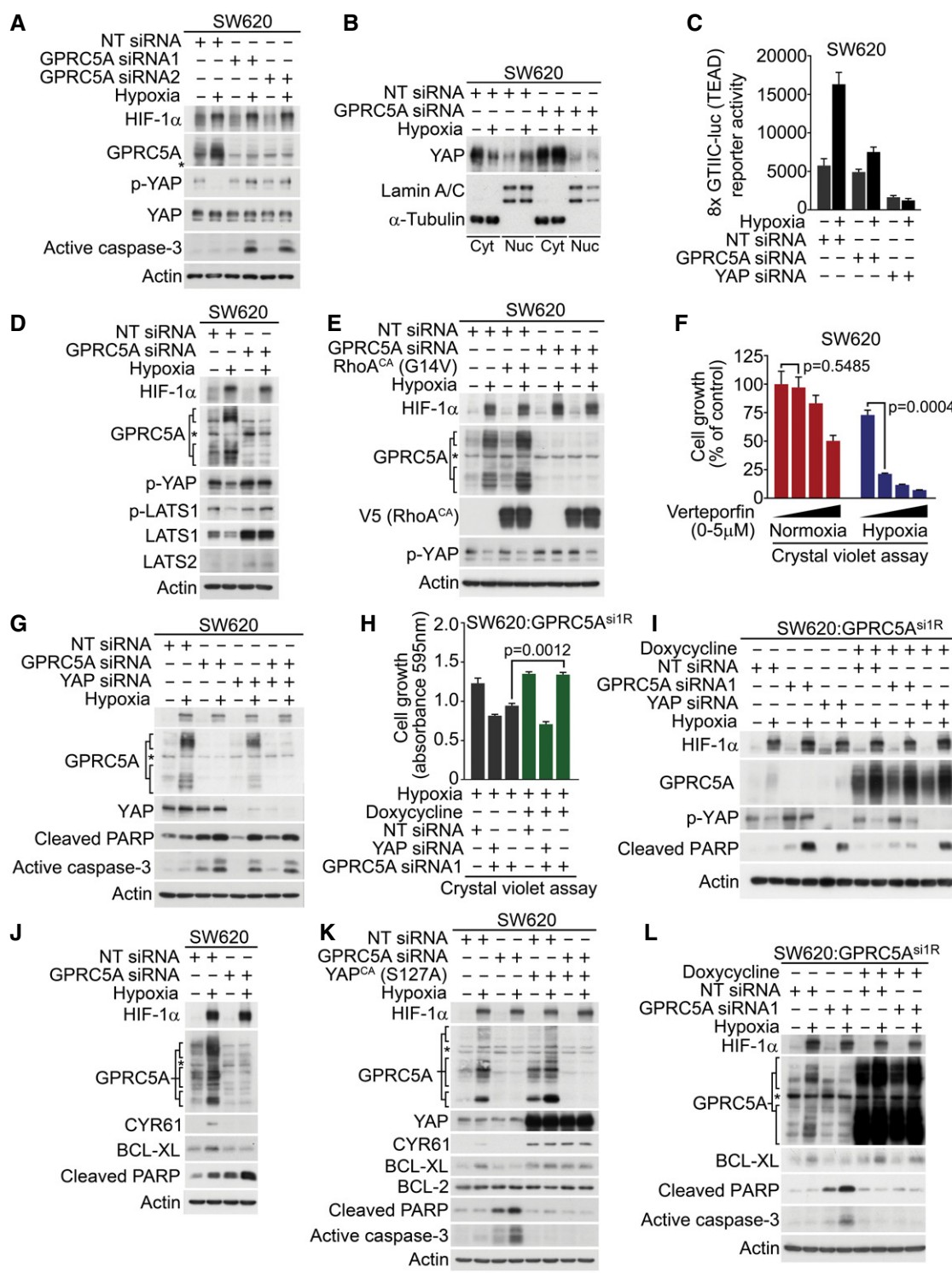

**Figure 4.**

GPRC5A depletion in hypoxia (Fig 3D). To accurately quantify the apoptosis resulting from GPRC5A depletion, we used the violet ratiometric membrane asymmetry probe and flow cytometry (which we validated and optimised using the apoptosis-inducing drug ABT-737 and caspase inhibitor QVD; Fig EV3D). In normoxia, GPRC5A depletion modestly increased apoptosis compared with non-targeting

siRNA control (7.82% ± 1.34 vs. 2.17% ± 0.23, ~ 3.6-fold increase), but this increased markedly in hypoxia (23.37% ± 2.06 vs. 1.84% ± 0.21; ~ 12.7-fold increase; Fig 3E). The pro-apoptotic effect of GPRC5A depletion in hypoxia was further validated using the caspase inhibitor QVD, which rescued the apoptotic phenotypes in both flow cytometry (Fig 3E) and Western blot (Fig 3F) analyses.

**Figure 4.  GPRC5A promotes hypoxic cell survival via a novel HIF-GPRC5A-YAP axis.**

A   Hypoxia-induced YAP stabilisation via Ser397 dephosphorylation was abrogated in GPRC5A-depleted cells.
B   Hypoxia-induced nuclear localisation of YAP was attenuated in GPRC5A-depleted cells.
C   Hypoxia stimulated TEAD activity (8× GTIIC-luc reporter), but this effect was reduced by GPRC5A depletion. A representative triplicate experiment is shown (*n* = 3).
D   Hypoxia reduced LATS activity and expression, but this was prevented by GPRC5A depletion.
E   Constitutively active RhoA (G14V) expression restored YAP stabilisation (Ser397 dephosphorylation) by hypoxia in GPRC5A-depleted cells.
F   The YAP/TEAD inhibitor verteporfin selectively inhibited cancer cell survival in hypoxia by crystal violet assay (*n* = 3 independent experiments).
G   YAP knockdown was sufficient to induce caspase-3 activation/PARP cleavage in hypoxia and was not further enhanced by GPRC5A depletion.
H   Crystal violet assays show that YAP was required downstream of GPRC5A to promote cell survival. GPRC5A-depleted cells were rescued by expression of an siRNA-resistant GPRC5A cDNA (GPRC5A$^{si1R}$), but this was prevented by co-depletion of YAP (*n* = 3 independent experiments).
I    Expression of an siRNA-resistant GPRC5A rescued the critical phenotypes of GPRC5A depletion. GPRC5A$^{si1R}$ expression prevented PARP cleavage in hypoxia as well as restoring hypoxia-induced YAP stabilisation (Ser397 dephosphorylation); these phenotypes were reversed by YAP depletion.
J    GPRC5A depletion attenuated hypoxia-induced BCL-XL expression.
K   Constitutively active YAP (S127A) expression induced BCL-XL expression and prevented caspase-3 activation/PARP cleavage by GPRC5A depletion in hypoxia.
L    GPRC5A$^{si1R}$ expression restored BCL-XL expression and prevented the appearance of cleaved caspase-3 induced by GPRC5A depletion in hypoxia.

Data information: Asterisks (*) indicate non-specific band. Level adjustments were made to images in Adobe Photoshop post-acquisition for clarity (equal changes applied to the entire image). Representative examples of *n* = 3 independent experiments are shown; data are presented as mean ± SEM. One-way ANOVA with Tukey's multiple comparisons test was carried out in (F and H).
Source data are available online for this figure.

## YAP is required downstream of GPRC5A for cancer cell survival in hypoxia

These data strongly indicate that GPRC5A protects tumour cells from apoptosis during hypoxia.

To gain a mechanistic insight into how GPRC5A might promote cell survival during hypoxia, we performed additional bioinformatic analysis on transcriptomics dataset GSE24551. KEGG pathway and gene ontology analyses revealed that *GPRC5A* mRNA expression strongly correlated with genes related to Hippo signalling, particularly in later stage tumours (Appendix Table S2). Since YAP is a major downstream regulator of the Hippo pathway and is required for the formation and growth of colorectal tumours (Rosenbluh *et al*, 2012; Zanconato *et al*, 2016), we examined its regulation by hypoxia and found increased total YAP protein in a panel of colorectal tumour cell lines (Fig EV4A). To determine whether there was a functional connection between GPRC5A and YAP in hypoxia, we examined the stability of YAP during hypoxia in the presence and absence of GPRC5A siRNAs. In control cells, YAP Ser397 phosphorylation decreased during hypoxia, indicating that hypoxia stabilises YAP (Fig 4A). Remarkably, however, we found that in GPRC5A-depleted cells, YAP phosphorylated at Ser397 persisted during hypoxia, suggesting that the stabilisation of YAP during hypoxia is dependent on GPRC5A (Fig 4A). To confirm this, we examined YAP nuclear localisation and transcriptional activity using the 8× GTIIC-luciferase (TEAD) reporter. Nuclear YAP levels increased during hypoxia, and this was abrogated by GPRC5A depletion (Fig 4B). Furthermore, GPRC5A depletion reduced hypoxia-induced TEAD activity (Fig 4C) and expression of established YAP target genes *AREG*, *CYR61*, *CTGF* and *BCL2L1* (Fig EV4B–E).

Having shown that GPRC5A depletion reduces YAP activity during hypoxia, we asked how GPRC5A might regulate YAP phosphorylation. The small GTPase RhoA has previously been reported to be a positive upstream regulator of YAP activity (Park *et al*, 2015) via its inhibition of LATS1/2 kinases downstream of Gα$_{12/13}$-coupled receptors (Yu *et al*, 2012). In line with decreased YAP Ser397 phosphorylation during hypoxia, we found that both activated (phosphorylated) LATS1 levels and total expression of LATS1/2 decreased during hypoxia (as reported previously for LATS2; Ma *et al*, 2015). However, the inhibitory effects of hypoxia on both LATS activity and expression were prevented in cells depleted of GPRC5A (Fig 4D). This suggests that GPRC5A depletion may stabilise LATS1/2 leading to increased YAP phosphorylation. In line with this, expression of a constitutively active (G14V) form of RhoA was sufficient to reverse the inhibitory phosphorylation of YAP caused by GPRC5A depletion in hypoxia (Fig 4E), and we saw no additional increase in YAP phosphorylation in hypoxic GPRC5A-depleted cells upon expression of a dominant negative (T19N) form of RhoA (Fig EV4F). Interestingly, we noticed that expression of constitutively active RhoA enhanced GPRC5A expression in hypoxia (Fig 4E), suggesting that activation of YAP may further increase GPRC5A expression via a positive feedback loop. These data suggest that RhoA is likely to act downstream of GPRC5A in hypoxia to promote YAP stabilisation and activity.

To determine whether YAP activation downstream of GPRC5A signalling has an anti-apoptotic role during hypoxia, it was first important to test whether hypoxic cancer cells require YAP for survival. To do this, we treated cells with verteporfin, a drug previously established to inactivate the YAP pathway by disrupting YAP/TEAD protein–protein interactions (Liu-Chittenden *et al*, 2012). As a proof of principle, hypoxic SW620 cells were profoundly more sensitive to verteporfin relative to normoxia controls (Fig 4F). We then asked whether YAP activation downstream of GPRC5A during hypoxia was necessary for cell survival. Depletion of both GPRC5A and YAP together revealed no additive effects on apoptosis phenotypes relative to depletion of either protein alone (Fig 4G), suggesting that YAP may be downstream of GPRC5A. To confirm this, we tested whether YAP deficiency failed to protect cells from apoptosis in a GPRC5A-dependent manner using GPRC5A$^{si1R}$-inducible cells. As expected, doxycycline-induced expression of GPRC5A$^{si1R}$ cDNA rescued the cell growth and apoptosis phenotypes caused by GPRC5A depletion in hypoxia (Fig 4H and I). Furthermore, GPRC5A$^{si1R}$ expression reversed the increased YAP Ser397 phosphorylation phenotype associated with GPRC5A depletion (Fig 4I). Strikingly, although GPRC5A$^{si1R}$ expression was able to rescue the cell growth and apoptosis phenotypes of GPRC5A depletion in hypoxia, these effects were reversed in the absence of YAP (Figs 4H

and I, and EV4F). These data indicate that YAP acts downstream of GPRC5A to promote cell survival during hypoxia.

We then asked how signalling downstream of the HIF-GPRC5A-YAP axis might protect hypoxic cancer cells from apoptosis. Recent reports have shown that *BCL2L1* (encoding BCL-XL) is an important anti-apoptotic downstream target gene of YAP (Imajo *et al*, 2012; Rosenbluh *et al*, 2012). We observed a marked upregulation of *BCL2L1* transcripts and BCL-XL protein during hypoxia, which was clearly absent in GPRC5A-depleted cells (Figs 4J and EV4C). To confirm that the apoptosis induced by GPRC5A depletion in hypoxia was contingent on a failure to activate YAP, we performed rescue experiments with a constitutively active YAP mutant (YAP[CA], S127A). Constitutively active YAP expression led to increased expression of BCL-XL (and CYR61) and prevented the induction of apoptosis by GPRC5A depletion in hypoxia (Fig 4K). Finally, we confirmed that BCL-XL is downstream of the GPRC5A-YAP axis by expressing GPRC5A[si1R], which restored BCL-XL expression and prevented apoptosis in hypoxic GPRC5A-depleted cells (Fig 4L). Overall, our findings reveal a novel HIF-GPRC5A-RhoA-YAP axis that is important for cancer cell survival during hypoxia (see synopsis model; Fig EV5).

# Discussion

The development of solid tumours is accompanied by the onset of hypoxia, making tumour cell evasion of apoptosis instigated by hypoxic stress a central step during tumorigenesis (Harris, 2002; Vaupel & Mayer, 2007; Hanahan & Weinberg, 2011; Semenza, 2012). For cells in developing tumours, hypoxia poses a selection pressure that drives the acquisition of malignant traits, leading to poor clinical outcomes and resistance to therapies (Harris, 2002; Bottaro & Liotta, 2003; Vaupel & Mayer, 2007; Wilson & Hay, 2011). Given hypoxia's impact in shaping the progression of many common solid tumours, understanding its influence on signalling pathways involved in tumorigenesis is of major importance (Wilson & Hay, 2011). While the HIF-driven transcriptional response is known to be important for hypoxic adaptation, key unanswered questions remain in understanding which genes downstream of HIFs are most critical for cell survival, and how HIFs cooperate with other important oncogenic pathways to orchestrate malignant behaviour. In addition, while hypoxia is widely thought to be a valid cancer-specific therapeutic target (Wilson & Hay, 2011), one reason it has not yet been fully exploited in the clinic is the lack of druggable molecules downstream of the HIF transcriptional response. Our work identifies GPRC5A as a previously unrecognised mediator of tumour cell survival specifically during hypoxia. This represents a new opportunity to exploit the notion of "conditional synthetic lethality", leveraging the cancer-associated nature of hypoxia to selectively target tumour cells for death.

Although *GPRC5A* was originally identified as a retinoic acid-inducible gene (Cheng & Lotan, 1998), the transcription factors responsible for its regulation have remained unclear. Our work indicates that HIFs are major regulators of GPRC5A expression. This could explain its low levels in many normal tissues, as most tissues are not hypoxic under normal physiological conditions (Wilson & Hay, 2011). Furthermore, our data provide a molecular basis to explain GPRC5A overexpression previously observed in solid tumours, such as pancreatic cancers (Zhou & Rigoutsos, 2014; Zhou

*et al*, 2016). GPRC5A may also be induced by oncogenic pathways upstream of HIFs such as RAS and PI3K pathways, or by other conditions that result in "pseudo-hypoxia" (Sabharwal & Schumacker, 2014). Curiously, while *Gprc5a* constitutive knockout mice (*Gprc5a*[−/−]) are reported to have a higher prevalence of spontaneous late-onset lung tumours than their wild-type littermates (Tao *et al*, 2007; Kadara *et al*, 2010), a recent report showed that the same *Gprc5a*[−/−] mice are actually resistant to inflammation-induced intestinal tumorigenesis, with dramatic reductions intestinal tumours ≥ 3 mm (Zhang *et al*, 2017). Based on our mechanistic findings, we speculate that without GPRC5A, such tumours would be compromised in their ability to grow due to a failure to engage the HIF-GPRC5A-YAP axis, thereby unable to survive hypoxic stress. This is further supported by emerging evidence that indicates pro-survival roles for YAP in oxygen-regulated processes in a variety of contexts (Ma *et al*, 2015; Santinon *et al*, 2016; Wang *et al*, 2017).

In our experiments, we found that activation of YAP by hypoxia required GPRC5A, which establishes YAP as a major downstream effector of HIF-driven GPRC5A-dependent cell survival. However, this raises an interesting question: if activation of YAP enables cell survival in hypoxia, why would HIFs—as the "primary sensors" of oxygen deprivation—not activate YAP directly, rather than via a cell surface GPCR? The answer to this may lie in GPRC5A's potential role as a "secondary metabolic sensor", providing an additional level of microenvironmental sensing downstream of HIFs that connects the extracellular milieu to appropriate intracellular responses. Indeed, Hippo–YAP signalling has recently been established as a critical signalling branch downstream of certain GPCRs in response to extracellular diffusible signals (Yu *et al*, 2012). Furthermore, while GPRC5A remains an orphan receptor for which no ligands have been identified, a study on the *Drosophila* GPRC5-family orthologue *BOSS* suggests that this family of receptors may be involved in sensing glucose to regulate sugar and lipid metabolism (Kohyama-Koganeya *et al*, 2008). Since hypoxia is known to trigger HIF-dependent expression of genes involved in maximising glucose uptake during glycolysis, and glycolysis has been reported to activate YAP (Enzo *et al*, 2015), GPRC5A may form part of the HIF-driven metabolic shift during hypoxia by acting as a sensor for extracellular cues to facilitate cellular adaptation and survival.

The discovery of the HIF-GPRC5A-YAP axis could have broad implications for targeted oncology, because hypoxia occurs during the development and progression of many common adult and childhood solid tumours. Furthermore, targeting GPRC5A may also provide a new means to inhibit tumour-specific YAP activity. Moreover, since hypoxia frequently occurs in the therapy-resistant regions of tumours where the utility of hypoxia-targeted drugs is limited, the development of inhibitors to GPRC5A may lead to novel cancer-selective drugs that could serve as adjuncts to conventional chemo- and radiotherapy. Given that the HIF and Hippo–YAP pathways play important physiological roles in normal development as well as during pathophysiological conditions, the crosstalk mediated by GPRC5A reported here has functional consequences outside its role in the context of tumorigenesis. For example, GPRC5A was recently found to be highly expressed at the leading edge of wounds (Aragona *et al*, 2017), so a more detailed understanding of its functions may have implications for regenerative medicine, as well as oncology.

# Materials and Methods

### Cell culture experiments

The human colorectal tumour cell lines Caco2, DLD1, HT29, HCT15, HCT116, LOVO, LS174T, RKO, SW480 and SW620 were obtained from the American Type Culture Collection (ATCC; Rockville, USA). The RG/C2 adenoma cell line was derived in this laboratory and was grown as described previously (Greenhough et al, 2010). Normoxia and hypoxia treatments were carried out in DMEM (Gibco 12491-015) containing 10% FBS, supplemented with penicillin (100 units/ml), streptomycin (100 μg/ml) and glutamine (4 mM). Dimethyloxalylglycine (DMOG) was from Sigma; verteporfin was from MedChemExpress.

### Hypoxia treatments

Hypoxia treatments were carried out at 37°C by flushing an InvivO$_2$ 300 hypoxia workstation/incubator (Ruskinn) with nitrogen to create an atmosphere containing 94% N$_2$, 5% CO$_2$ and 1% O$_2$. For hypoxic treatment of zebrafish embryos (up to 5 days post-fertilisation), 5% O$_2$ was used (Santhakumar et al, 2012).

### SILAC labelling and proteomics

SILAC reagents were from Thermo Fisher Scientific; SILAC medium and dialysed FBS were from Gibco. SW620 cells were grown in the SILAC medium for at least six doublings to achieve full labelling. Whole cell lysates were subjected to LC-MS/MS analysis on an LTQ Orbitrap Velos mass spectrometer (Thermo) as described below.

### LC-MS analysis

SILAC-labelled samples were pooled and ran on a 10% SDS–PAGE gel, and the gel lane was cut into 10 equal slices. Each slice was subjected to in-gel tryptic digestion using a DigestPro automated digestion unit (Intavis), and the resulting peptides were fractionated using an UltiMate 3000 nano-LC system in line with an LTQ Orbitrap Velos mass spectrometer (Thermo). Briefly, peptides in 1% (vol/vol) formic acid were injected onto an Acclaim PepMap C18 nano-trap column (Thermo). After washing with 0.5% (vol/vol) acetonitrile, 0.1% (vol/vol) formic acid peptides were resolved on a 250 mm × 75 μm Acclaim PepMap C18 reverse phase analytical column (Thermo) over a 150 min organic gradient, using seven gradient segments (1–6% solvent B over 1 min, 6–15% B over 58 min, 15–32% B over 58 min, 32–40% B over 5 min, 40–90% B over 1 min, held at 90% B for 6 min and then reduced to 1% B over 1 min) with a flow rate of 300 nl/min. Solvent A was 0.1% formic acid, and Solvent B was aqueous 80% acetonitrile in 0.1% formic acid. Peptides were ionised by nano-electrospray ionisation at 2.1 kV using a stainless-steel emitter with an internal diameter of 30 μm (Thermo) and a capillary temperature of 250°C. Tandem mass spectra were acquired using an LTQ Orbitrap Velos mass spectrometer controlled by Xcalibur 2.0 software (Thermo) and operated in data-dependent acquisition mode. The Orbitrap was set to analyse the survey scans at 60,000 resolution (at m/z 400) in the mass range m/z 300–2,000 and the top ten multiply charged ions in each duty cycle selected for MS/MS in the LTQ linear ion trap. Charge state filtering, where unassigned precursor ions were not selected for fragmentation, and dynamic exclusion (repeat count, 1; repeat duration, 30 s; exclusion list size, 500) were used. Fragmentation conditions in the LTQ were as follows: normalised collision energy, 40%; activation q, 0.25; activation time, 10 ms; and minimum ion selection intensity, 500 counts.

### Proteomics data analysis

Raw data files were processed and quantified using Proteome Discoverer software v1.4 (Thermo) and searched against the UniProt Human database (122,604 entries) using the SEQUEST algorithm. Peptide precursor mass tolerance was set at 10 ppm, and MS/MS tolerance was set at 0.8 Da. Search criteria included carbamidomethylation of cysteine (+57.0214) as a fixed modification and oxidation of methionine (+15.9949) and appropriate SILAC labels ($^2$H$_4$-Lys, $^{13}$C$_6$-Arg for duplex and $^{13}$C$_6$$^{15}$N$_2$-Lys and $^{13}$C$_6$$^{15}$N$_4$-Arg for triplex) as variable modifications. Searches were performed with full tryptic digestion, and a maximum of one missed cleavage was allowed. The reverse database search option was enabled, and all peptide data were filtered to satisfy false discovery rate (FDR) of 5%.

### Western blotting

Western blot analysis was performed as described previously (Petherick et al, 2013) using the following antibodies: GPRC5A (1:2,000, CST, 12968), β-actin (1:10,000, Sigma, A5316), HIF-1α (1:1,000, BD, 610959), HIF-1β (1:1,000, BD, 611078), HIF-2α (1:1,000, CST, 7096), PLOD2 (1:1,000, R&D, MAB4445), CA9 (1:5,000, Novus, NB100-417), cleaved PARP (1:20,000, Abcam, ab32064), active caspase-3 (1:1,000, CST, 96645), p-YAP S397 (1:5,000, CST, 13619), YAP (1:5,000, CST, 14074), BCL-XL (1:1,000, BD, 556361), BCL-2 (1:200, Santa Cruz, SC-509), V5-tag (1:2,000, CST, 13202), CYR61 (1:2,000, Santa Cruz, SC-374129), RhoA (1:2,000, CST, 2117), lamin A/C (1:10,000, Sigma, 4C11) and α-tubulin (1:10,000, Sigma, T6199). Cells were washed with ice-cold PBS and lysed on ice for 10 min with Cell Signaling Technology lysis buffer (9803) supplemented with protease inhibitors and sonicated briefly. Equal protein concentrations were resolved using sodium dodecyl sulphate–polyacrylamide gel electrophoresis (SDS–PAGE) and transferred to an Immobilon-P polyvinylidene difluoride membrane (Millipore). For GPRC5A Western blots, samples were not boiled.

### Bioinformatics

Analysis of dataset GSE24551 (Sveen et al, 2011) was carried out using R2: Genomics Analysis and Visualization Platform (http://r2.amc.nl). KEGG pathway and gene ontology (GO) analyses were carried out using GPRC5A as the source gene in tumour samples (n = 320). Gene set enrichment analysis (GSEA) datasets used were Semenza_HIF1_Targets (M12299) Broad_Hallmark_Hypoxia (M5891) and are available from http://software.broadinstitute.org.

### Immunohistochemistry

Tissue blocks (formalin-fixed, paraffin-embedded tissue) were obtained from the archives of the Department of Histopathology at

the Bristol Royal Infirmary, Bristol, England, UK, after approval from the local research ethics committee (REC reference: E5470). These were sectioned by the Histology Services Unit at University of Bristol. For antibody validation, following siRNA knockdown of CA9 and GPRC5A, cell lines were formalin-fixed and paraffin-embedded as described previously (Greenhough et al, 2010). Antibodies were used at a 1:800 dilution. This study involves only using archival, anonymised tissues blocks, held prior to 1 September 2006. From the Human Tissue (HT) Act code of practice for research, the consent requirements of the HT Act are not retrospective. This means that legally it is not necessary to seek consent under the HT Act to store or use an "existing holding" for a scheduled purpose. An existing holding is a material from the living or deceased that was already held at the time the HT Act came into force on 1 September 2006.

### Transfection, siRNA and 8× GTIIC-luc reporter assays

Transfections were performed using Lipofectamine 2000 or RNAiMax (Invitrogen) in Opti-MEM (Gibco). Cells were transfected twice with siRNA (20 nM) in a 48-h period to maximise knockdown; siRNAs to human GPRC5A, HIF1A, HIF2A/EPAS1, HIF1B/ARNT, or validated non-targeting negative control are detailed in Appendix Table S3. Briefly, cells were transfected overnight; normal growth medium was replaced the next day followed by a further transfection overnight. Twenty-four hours later, cells were treated as indicated in figure legends. 8× GTIIC-luc (YAP/TEAD) reporter activity was determined using a dual-luciferase reporter system (Promega) as described previously (Kimura et al, 2016). Constructs and methods for adenoviral expression of active YAP (S127A), active RhoA (G14V) and dominant negative RhoA (T19N) have been described previously (Kimura et al, 2016).

### Cloning and establishment of stable cell lines

Site-directed mutagenesis (Genewiz) was used to generate GPRC5A cDNA resistant to GPRC5A siRNA sequence 1 (termed GPRC5A[si1R]). The GPRC5A open reading frame was amplified from GPRC5A[si1R] using primers containing AvrII and BsrgI restriction sites (Appendix Table S4), cloned into pCW57-GFP-2A-MCS (a kind gift from Adam Karpf, Addgene plasmid #71783) and verified by DNA sequencing (Source Biosciences). SW620 cells were lentivirally infected and puromycin-selected (15 μg/ml) for 7 days. To obtain near 100% expressing cells, SW620:GPRC5A[si1R] were treated with doxycycline (2.5 μg/ml) for 48 h prior to flow sorting based on medium/high TurboGFP expression (self-cleaving from the pCW57-GFP-2A-MCS plasmid) using a BD Influx cell sorter (Becton-Dickinson).

### Immunofluorescence

Confocal analysis of immunofluorescence was carried out as described previously (Petherick et al, 2013). Hypoxia-treated cells grown on coverslips were fixed with 4% paraformaldehyde (with 0.1% Triton X-100) and stained with GPRC5A (1:200, CST, 12968) and secondary antibodies (1:2,000) prior to DAPI nuclear staining and mounting slides. Images were processed in Adobe Photoshop (as described in corresponding legends).

### Quantitative reverse transcription-polymerase chain reaction (qRT–PCR)

Following treatment, RNA was extracted using TRI reagent (Sigma), chloroform and isopropanol. After purification using the TURBO DNase kit (Ambion), complementary DNA was produced using the MMLV reverse transcriptase kit (Promega). qRT–PCR was performed using a SYBR Green PCR kit (Qiagen) in a Stratagene MX3005P qPCR cycler (La Jolla). A list of primers used in the study is provided in Appendix Table S4.

### Chromatin immunoprecipitation (ChIP)

ChIP was carried out as described previously (Petherick et al, 2013). Briefly, 1% formaldehyde-fixed chromatin from nuclear lysates was sheared to a 500 bp average by sonication (Diagenode Bioruptor), pre-cleared and subject to immunoprecipitation overnight at 4°C with antibodies to normal mouse IgG (8 μg/ml, Millipore, 12-371), RNA polymerase II (8 μg/ml, Millipore) and HIF-1α (8 μg/ml, BD, 610959). Samples were incubated for a further 1 h at 4°C with Protein G Magnabeads (Invitrogen) and processed using a DynaMag-2 magnetic particle separator (Invitrogen).

### Flow cytometry (violet ratiometric membrane asymmetry assay)

Following treatments, cells were washed with PBS and incubated with F2N12S at 200 nM and SYTOX AADvanced dead cell stain (A35137; Life Technologies) for 5 min at room temperature (as per the manufacturer's instructions). Live, dead and apoptotic cells were detected using a BD LSR II flow cytometer (Becton-Dickinson) and FACSDiva software. 30,000 events were measured, and the data were analysed using FlowJo v10 analysis software (Tree Star, Inc.).

### Crystal violet cell growth assays

Cells were seeded into 96-well plates (in sextuplicate per condition). Following treatments, cells were fixed for 10 min in 4% paraformaldehyde, stained with in 0.5% crystal violet solution (Sigma) and solubilised in 1% SDS solution before reading absorbance values at 595 nm.

### Mouse and zebrafish experiments

All experiments were conducted with approval from the local ethical review committee at the University of Bristol and in accordance with the UK Home Office regulations (Guidance on the Operation of Animals, Scientific Procedures Act, 1986). Mice (Mus musculus) were from a mixed C57Bl6/J background and aged 6–10 weeks when induced. Both sexes were used. Mice contained the following inducible genetic modifications (floxed alleles): Villin-CreERT2 Apc[fl/fl] (Shibata et al, 1997; el Marjou et al, 2004) and Villin-CreERT2 Apc[fl/fl]; Hif1a[fl/fl] (Ryan et al, 1998) and induced with tamoxifen at 6–10 weeks of age. The following induction regimes were used: Apc[fl/fl] mice 80 mg/kg tamoxifen (intraperitoneal) on day 0 and day 1; samples were generated on day 4 post-induction. Total RNA was isolated from small intestinal tissue to generate cDNA as described previously (Huels et al, 2015). Zebrafish (Danio rerio), home bred, up to 5 days post-fertilisation larvae (therefore gender

## The paper explained

### Problem

The development of many common cancers is accompanied by the onset of hypoxia (reduced tissue oxygen levels) because tumours often outgrow their blood supply. When faced with oxygen deprivation, cancer cells are forced to adapt—this in turn drives malignant progression, metastases and drug resistance. Because hypoxia is cancer-specific in nature, understanding how cancer cells adapt to hypoxia may lead to therapies that can selectively kill cancer cells that rely on hypoxia-induced signalling for their survival, while sparing normal tissue. Although it is known that hypoxia-inducible factors (HIFs) "switch on" proteins that help cells adapt to hypoxia, identifying those most important for hypoxic cancer cell survival that are also "druggable" remains an unmet challenge.

### Results

Using multidisciplinary approaches *in vitro* and *in vivo*, we have characterised a previously unrecognised mechanism that facilitates cancer cell adaptation and survival in hypoxia. By analysing the entirety of proteins whose levels are changed following cancer cell growth in low oxygen, we identified elevated expression of a G protein-coupled receptor, called GPRC5A. Using bioinformatics, we found that high levels of *GPRC5A* mRNA correlate with hypoxia gene signatures and poor survival outcomes in colorectal cancer patients. Mechanistic studies revealed that GPRC5A promotes the activity of the oncoprotein YAP, a signal transducer from the Hippo signalling pathway with established roles in cancer development. YAP in turn switches on a protein, BCL-XL, which enables cancer cells to tolerate and survive low oxygen-induced stress. We term this cancer cell survival pathway the "HIF-GPRC5A-YAP axis".

### Impact

Hypoxia has been proposed as one of the best validated cancer-selective targets not yet exploited in the clinic, and one reason for this is the lack of "druggable" molecules downstream of the HIF transcriptional response. GPCRs are considered to be amongst the best drug targets for many diseases, and therefore, GPRC5A represents a new opportunity to exploit the notion of "conditional synthetic lethality"—leveraging the cancer-associated nature of hypoxia to selectively target tumour cells for death.

N/A), were fed paramecia/rotifer chow and maintained in a 14:10-light/dark cycle at 28.5 Celsius. The transgenic zebrafish line Tg(*fli1*:eGFP) was crossed onto the *vhl*[hu2117] mutant background (Watson *et al*, 2013) as described previously (van Rooijen *et al*, 2009). The transgenic zebrafish line Tg(*fli1*:eGFP) was crossed onto the *vhl*[hu2117] mutant background (Watson *et al*, 2013) as described previously (van Rooijen *et al*, 2009).

### Statistical analyses

For animal studies, where appropriate, the Experimental Design Assistant (provided by the National Research Centre for the Replacement Refinement and Reduction of Animals in Research) was used to calculate sample size https://www.nc3rs.org.uk/experimental-design-assistant-eda. No animals were excluded from the analysis. No steps were taken to randomise sample allocation, and no randomisation was used. There were no scoring experiments and no blinding was done. The Shapiro–Wilk normality tests were carried out for normal distribution. Standard deviation and standard error of the mean where appropriate are indicated. Analyses using *t*-tests (one-sample and Student's) and ANOVA were performed as indicated in figure legends (*P*-values are indicated). Data are expressed as the mean $\pm$ SD or $\pm$SEM (as indicated in figure legends).

## Data availability

Mass spectrometry proteomics data are available in the following database: ProteomeXchange Consortium via PRIDE PXD009971 (https://www.ebi.ac.uk/pride).

**Expanded View** for this article is available online.

## Acknowledgements

We thank all members of the Williams, Martin and Malik laboratories for their valuable support and discussion. We would like to thank Andrew Herman and Lorena Sueiro Ballesteros and the University of Bristol Flow Cytometry Facility for cell sorting and analysis; Debi Ford and Debbie Martin and the Histology Services Unit for help with immunohistochemistry; Marianna Szemes for help with bioinformatics/R2 analysis; Anne Ridley and Harry Mellor for RhoA antibodies and for useful discussions. We thank the University of Bristol Proteomics Facility, and the Wolfson Foundation for supporting the Wolfson Bioimaging Facility at the University of Bristol. This work was funded by a Cancer Research UK Programme Grant to ACW and CP (C19/A11975; AG, TJC, ACW and CP) a University of Bristol studentship (to CB), a BBSRC grant to KM (BB/P008232/1; AG and KM) the Citrina Foundation (AG, TJC, ACW and CP) and the John James Bristol Foundation (AG, TJC, ACW and CP). PM is supported by a Wellcome Trust Investigator Award (WT097791/Z11/Z; DBG and PM) and a Cancer Research UK Programme Grant (C20590/A15936). KM is supported by a Cancer Research UK grant (A12743/A21046). OJS is supported by Cancer Research UK grants (C7932/A25045, C596/A17196, A12481 and A21139) and an ERC starting grant (COLONCAN/311301; OJS and DG).

## Author contributions

AG, CP and ACW conceived the study. AG coordinated the project. AG designed the experiments and with CB conducted most of the experiments (AG and CB made equal experimental contributions). AG and DBG designed and performed zebrafish experiments. DG carried out mouse experiments and AG performed the analysis. AG and KM designed and performed cloning experiments. AG and KM performed bioinformatics. AG and KJH conducted the SILAC-based LC-MS/MS proteomics experiments. MB developed and produced adenoviral constructs. TJC provided technical support. AG, CB, KJH, DBG, DG, KM, CP and ACW collected and analysed data. OJS was responsible for and supervised the mouse studies. PM was responsible for and supervised the zebrafish studies. AG wrote the manuscript. KM, PM and ACW helped to write the manuscript. AG, KM, PM and ACW reviewed, edited and revised the manuscript. All co-authors read and gave input to help improve the manuscript.

## Conflict of interest

The authors declare that they have no conflict of interest.

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
