## [Review Process File · EMBO Molecular Medicine]

Cancer cell adaptation to hypoxia involves a HIF-GPRC5A-YAP axis

Alexander Greenhough, Clare Bagley, Kate J. Heesom, David B. Gurevich, David Gay, Mark Bond, Tracey J. Collard, Chris Paraskeva, Paul Martin, Owen J. Sansom, Karim Malik & Ann C. Williams

Review timeline:

Submission date:	29 November 2017
Editorial Decision:	22 January 2018
Revision received:	29 May 2018
Editorial Decision:	05 July 2018
Revision received:	26 July 2018
Accepted:	27 July 2018

Editor: Céline Carret

Transaction Report:

1st Editorial Decision

22 January 2018

Thank you for the submission of your manuscript to EMBO Molecular Medicine. We have now heard back from the two referees whom we asked to evaluate your manuscript.

You will see from the comments pasted below, that both referees find the study to be of great interest. However, both request additional mechanistic insight (especially regarding the YAP link), further controls needed to quantify the data, provide cell growth experiment and more convincing siRNA experiments.

Overall, we would welcome the submission of a revised version within three months for further consideration and would like to encourage you to address all the criticisms raised as suggested to improve conclusiveness and clarity. Please note that EMBO Molecular Medicine strongly supports a single round of revision and that, as acceptance or rejection of the manuscript will depend on another round of review, your responses should be as complete as possible.

I look forward to receiving your revised manuscript.

***** Reviewer's comments *****

Referee #1 (Comments on Novelty/Model System for Author):

The paper identifies an orphan GPCR (GPRC5A) as a direct HIF target in colorectal epithelial and cancer cells, and makes a reasonable, if incomplete, case for a mechanism by which GPRC5A regulates apoptosis. It's a new, if not earth-shattering, set of observations, but does identify a new mechanism by which tumor hypoxia may regulate tumor progression.

Referee #1 (Remarks for Author):

This report describes the identification of the orphan G-protein coupled receptor GPRC5A as a HIF target gene in colorectal epithelial cells *in vitro* and *in vivo*, and its apparent role in suppressing cancer cell apoptosis by promoting YAP-dependent induction of Bcl-XL expression. The authors used a SILAC-based proteomics approach to identify hypoxia-induced proteins in colorectal cancer (CRC) cells, and in addition to the expected targets, discovered GPRC5A. This is an interesting story, and contributes a new potential mechanism by which hypoxia and HIFs regulate tumor progression.

In general, most of the authors' data are solid and convincing, and support the notion that GPRC5A is a direct HIF target in CRC cells and normal colonic epithelial cells (Figures 1 and 2). The use of both murine and zebrafish *in vivo* models is a strength of the paper. However, although the data in Figure 3 are consistent with the argument that HIF-induced GPRC5A expression reduces apoptosis in a YAP-dependent mechanism, additional data are required to make that argument robust.

Major concerns include:

1. Immunoblots showing PARP and caspase 3 cleavage don't give an accurate assessment of the degree to which apoptosis is being altered. Flow cytometric analysis using Annexin 5 (or other markers) is needed to quantify the percentage of cells are responding to manipulation of GPRC5A expression.
2. I have some concerns regarding the GPRC5A immunoblots and siRNA experiments. Given that the GPRC5A antibody shows multiple bands, the nature of which is not clear and could be discussed, it's not always obvious which of the many bands is being shown in Figures 1F, 1G, 1H, and throughout Figure 3. While that's easily addressable, the bigger issue is that the multiple siRNA constructs in Figure S2 produce only modest apparent knockdown (again - which bands?), none of which appear as convincing as the knockdown in Figure 1B. Given this variability and complexity, the authors should rule out potential off-target effects by re-expressing an siRNA-resistant GPRC5A cDNA in knockdown cells (or, preferably, CRISPR-Cas9 KO cells) to show that it can rescue the critical phenotypes. Otherwise, the specter of possible off-target effects will remain.
3. Expression of the constitutively active YAP protein simply shows that YAP activity is sufficient to block apoptosis, but doesn't necessarily place YAP downstream of GPRC5A. Demonstrating that YAP deficiency fails to protect cells from apoptosis under hypoxia, in a GPRC5A-dependent manner, would be more convincing (sufficiency vs. necessity).

Minor points:

1. The potential mechanisms by which GPRC5A regulates YAP phosphorylation are important to discuss, despite being beyond the scope of this particular paper. Some discussion of this point is needed.
2. Any thoughts on why HIF1 and HIF2 both regulate GPRC5A in cell lines, but only HIF1 does so *in vivo*?
3. Higher magnification images of Figure 2C are needed to show sufficient detail.
4. The data in Figure 2G-I showing a correlation between hypoxia, GPRC5A expression, and CRC patient survival are of somewhat limited value. The Kaplan-Meier curves may be particularly

misleading, as GPCR5A expression in this context could simply be a surrogate for general HIF activation (see Kaelin, WG, (2017) Nature Rev Cancer 17: 425), as opposed to reflecting any functional role. This point should probably be acknowledged.

Referee #2 (Remarks for Author):

Greenough et al identified GPCR5A as a hypoxia inducible gene in proteomic analysis of colonic cancer cell line. They showed that GPCR5A is a direct and shared target of HIF1a/2a. The authors showed that GPCR5A overexpression protected cells from hypoxia-induced apoptosis (by Caspase WB). Bioinformatic analysis of colorectal cancer datasets revealed indicated that GPCR5A expression correlated with YAP target gene signatures. Departing from this point the authors conduct a series of experiments in which they show that hypoxia decreases YAP S397 and activates YAP target genes, one of which is BCL2L1, in a GPCR5A-dependent way. Expression of the degen-resistant YAP S127A rescues caspase activation in GPCR5A knock down hypoxic cells.

The manuscript presents a very interesting observation linking hypoxic survival to GPCR5A through YAP activation. The experiments are technically superb and the conclusions are supported by the data. This is an important link between hypoxia and YAP function which is novel.

There are specific aspects of this work that need to be clarified.

1) The paper does not address at all the mechanism by which GPCR5A alters YAP phosphorylation. Hypoxia was shown to activate YAP by degrading LATS2, in a SIADH2-dependnet way. Does GPCR5A signals through LATS1/2? Does it prevent LATS2 degradation? Is this done through Rho GTPases and if yes which one (RhoA/B/C). There is a need for biochemical experiments to address this mechanistic issue.

2) The effect of GPCR5A on cell growth during hypoxia is inferred, based on the expression of cleaved PARP or activated caspase. There are NO actual cell growth data (crystal violet assays) with cells in which GPCR5A and/or YAP are manipulated in ways similar to the ones that produced the caspase changes. This is an important detail that will confirm the biochemical observation transates into active cell growth differences.

3) The authors should test weather inactivation of YAP pathway in cells growing in hypoxia is sufficient to promote apoptosis.

4) Is non-hypoxic overexpression of GPCR5A sufficient to increase YAP S397 phosphorylation?

1st Revision - authors' response

29 May 2018

***** Reviewer's comments *****

Referee #1 (Comments on Novelty/Model System for Author):

The paper identifies an orphan GPCR (GPCR5A) as a direct HIF target in colorectal epithelial and cancer cells, and makes a reasonable, if incomplete, case for a mechanism by which GPCR5A regulates apoptosis. It's a new, if not earth-shattering, set of observations, but does identify a new mechanism by which tumor hypoxia may regulate tumor progression.

Referee #1 (Remarks for Author):

This report describes the identification of the orphan G-protein coupled receptor GPCR5A as a HIF target gene in colorectal epithelial cells in vitro and in vivo, and its apparent role in suppressing cancer cell apoptosis by promoting YAP-dependent induction of Bcl-XL expression. The authors used a SILAC-based proteomics approach to identify hypoxia-induced proteins in colorectal cancer (CRC) cells, and in addition to the expected targets, discovered GPCR5A. This is an interesting

story, and contributes a new potential mechanism by which hypoxia and HIFs regulate tumor progression.

In general, most of the authors' data are solid and convincing, and support the notion that GPRC5A is a direct HIF target in CRC cells and normal colonic epithelial cells (Figures 1 and 2). The use of both murine and zebrafish *in vivo* models is a strength of the paper. However, although the data in Figure 3 are consistent with the argument that HIF-induced GPRC5A expression reduces apoptosis in a YAP-dependent mechanism, additional data are required to make that argument robust.

We were pleased to read that Referee #1 found our story interesting and that they found most of our data to be solid and convincing. We have been able to address all of Referee #1's major and minor concerns. In doing so, we have included a significant amount of new data to increase the robustness of our conclusions (particularly surrounding the GPRC5A-YAP link).

Major concerns include:

1. Immunoblots showing PARP and caspase 3 cleavage don't give an accurate assessment of the degree to which apoptosis is being altered. Flow cytometric analysis using Annexin 5 (or other markers) is needed to quantify the percentage of cells are responding to manipulation of GPRC5A expression.

As requested by the referee, to accurately assess the degree to which apoptosis is being altered in response to hypoxia and GPRC5A depletion, we have complemented our cleaved PARP and caspase 3 immunoblots with flow cytometric analysis. We used an established (similar) alternative to Annexin 5, the violet ratiometric membrane asymmetry probe F2N12S/dead cell apoptosis assay (Thermo Scientific Catalogue number A35137) as described in Shynkar V et al, Fluorescent biomembrane probe for ratiometric detection of apoptosis. J Am Chem Soc (2007) 129: 2187-93. PMID: 17256940.

The new data are shown in Figure 3F (and Figure EV3D) where we quantify the degree to which apoptosis is altered by GPRC5A depletion in normoxia and hypoxia. Furthermore, as in our immunoblotting assays for cleaved PARP and caspase 3, the caspase inhibitor QVD reversed the effect of GPRC5A depletion on apoptosis (Figure 3E and 3F). In addition, as requested, using this assay we were also able to quantify the percentage of cells responding to manipulation of GPRC5A expression (live, apoptotic and dead, Figure 3F, lower panels).

2. I have some concerns regarding the GPRC5A immunoblots and siRNA experiments. Given that the GPRC5A antibody shows multiple bands, the nature of which is not clear and could be discussed, it's not always obvious which of the many bands is being shown in Figures 1F, 1G, 1H, and throughout Figure 3.

We agree that we could have made our presentation of the GPRC5A immunoblots clearer and we apologise for the lack of clarity. We have now included uncropped blots for GPRC5A in all figures where GPRC5A is shown (Figures 1B, 1C, 1E, 1F, 1G, 1H, 3A, 3B, 3D, 3E, 4A, 4B, 4D, 4E, 4G, 4I, 4J, 4K, 4L). In addition, and in line with EMBO's recommended policy, we will also make the source data (i.e. the uncropped, unprocessed films with molecular weight annotation) for immunoblots available to readers in the event of acceptance.

Detection of GPCRs by immunoblotting is known to be challenging, and protein lysates for GPRC5A immunoblots must be treated sensitively and prepared without boiling (to avoid GPCR aggregation). It should be noted that the appearance of GPRC5A's multiple bands by immunoblot can vary – this is likely due to GPCR dimerization and post-translational modifications (but also due to other factors, for example, variations in the percentage polyacrylamide gel used). Using our protocol, we consistently detect bands of ~30kDa, ~40kDa and ~80kDa (potential homodimers resistant to SDS-PAGE) in all cell lines tested: importantly, these are the bands that are sensitive to GPRC5A siRNA depletion. The antibody we use (Cell Signaling Technology Rabbit mAb #12968) also detects a band of ~60kDa; this is a non-specific band (i.e., it is insensitive to GPRC5A siRNA) and as such we have labelled this with an asterisk in all GPRC5A blots (which serves as a useful reference point as an approximate molecular weight marker).

To make this clear, we have included the following discussion in the body text:

In line with our proteomics data, western blotting confirmed GPRC5A to be induced by hypoxia (Fig 1B), apparent as a series of bands [likely due to dimerization and post-translational modifications (Zhou & Rigoutsos, 2014)] that we verified the identity of using GPRC5A siRNA (Fig 1C, note the non-specific ~60kDa band henceforth marked with an asterisk).

While that's easily addressable, the bigger issue is that the multiple siRNA constructs in Figure S2 produce only modest apparent knockdown (again - which bands?), none of which appear as convincing as the knockdown in Figure 1B.

We have now replaced Figure S2 with a new figure (Figure 3B) showing three independent siRNA sequences that knockdown GPRC5A protein to a similar extent. Furthermore, these siRNAs each produced similar hypoxia-specific increases in caspase-3 activation and PARP cleavage and similar effects on cell growth/survival by crystal violet assay (shown in Figure 3C).

Given this variability and complexity, the authors should rule out potential off-target effects by re-expressing an siRNA-resistant GPRC5A cDNA in knockdown cells (or, preferably, CRISPR-Cas9 KO cells) to show that it can rescue the critical phenotypes. Otherwise, the specter of possible off-target effects will remain.

We agree with the referee that this was an important omission. Therefore, to rule out potential off-target effects of siRNA, we designed and generated a codon-faithful (i.e., by synonymous mutations) GPRC5A cDNA construct resistant to GPRC5A siRNA#1 termed GPRC5A^{siR} (detailed in Figure 3D and EV3A-C). We cloned this cDNA into the doxycycline-inducible lentiviral overexpression construct pCW57-GFP-2A-MCS (a kind gift from Adam Karpf, Addgene Plasmid #71783). SW620 cells were transduced with lentivirus and selected with puromycin to generate stably transduced cells. To ensure that ~100% of cells were carrying the construct, we used flow cytometry to obtain a pure population of cells with GFP expression (following 48 hours growth in the presence of doxycycline) (Figure 3D and EV3A-C). Note that this plasmid produces a separate turbo GFP (not a fusion protein). These cells were then used to perform siRNA rescue experiments.

As shown in Figure 3D, knockdown of GPRC5A (with GPRC5A siRNA1) led to increased expression of apoptotic markers cleaved PARP and caspase-3 in hypoxic cells (consistent with our prior findings). Crucially, these phenotypes were rescued in the presence of doxycycline-induced GPRC5A^{siR}. As well as rescuing the appearance of apoptotic markers, doxycycline-induced expression of GPRC5A^{siR} also rescued the effect of GPRC5A depletion cell growth and survival (see Figure 4H). Furthermore, GPRC5A^{siR} expression also rescued the effect of GPRC5A depletion on YAP Ser397 phosphorylation (reversing the increased phosphorylation that occurs upon GPRC5A depletion in hypoxia, see Figure 4I). Similarly, the prevention of BCL-XL upregulation by GPRC5A depletion in hypoxia was rescued by expression of GPRC5A^{siR} (shown in Figure 4L).

Taken together with results from three independent GPRC5A siRNAs on apoptotic markers and cell growth/survival in hypoxia (Figure 3B and 3C), these results strongly suggest that the critical phenotypes observed with siRNA-mediated knockdown of GPRC5A result from 'on target' effects.

3. Expression of the constitutively active YAP protein simply shows that YAP activity is sufficient to block apoptosis, but doesn't necessarily place YAP downstream of GPRC5A. Demonstrating that YAP deficiency fails to protect cells from apoptosis under hypoxia, in a GPRC5A-dependent manner, would be more convincing (sufficiency vs. necessity).

We have added additional data to address this point. We show that YAP deficiency (genetic depletion using siRNA) or inhibition of YAP signalling (using the established YAP/TEAD inhibitor Verteporfin) increases apoptosis markers and reduces cell growth/survival preferentially in hypoxia (Figures 4F and 4G). In addition, we observe no further increases in cleaved PARP and caspase-3 when both GPRC5A and YAP are depleted together (versus

depletion of either protein alone). This indicates that depletion of either protein is sufficient to promote apoptosis in hypoxia and suggests they share common mode of action (Figure 4G). To address the referee's point more thoroughly, we show that doxycycline-induced expression of GPRC5A^{siIR} rescues the inhibitory effect of GPRC5A depletion on cell growth/survival in hypoxia, but that this effect is abolished by co-depletion of YAP (Figure 4H). Finally, we show that the ability of GPRC5A^{siIR} to prevent the appearance of apoptotic markers in GPRC5A depleted cells requires YAP, since depleting YAP in this context resulted in the re-appearance of apoptotic markers in hypoxic GPRC5A^{siIR} expressing cells (Figure 4I and 4L). Taken together with our data showing that YAP activity is sufficient to block apoptosis in hypoxic GPRC5A depleted cells (Figure 4K), our data strongly suggest that GPRC5A protects cells from apoptosis during hypoxia via YAP.

Minor points:

1. The potential mechanisms by which GPRC5A regulates YAP phosphorylation are important to discuss, despite being beyond the scope of this particular paper. Some discussion of this point is needed.

Although referee #1 suggests the mechanistic details linking GPRC5A to YAP are beyond the scope of our paper, given that referee #2 raised specific points related to this part of the study, we have performed additional experiments and added data indicating that GPRC5A signals to YAP in hypoxia via RhoA-LATS1/2 (please see the response to referee #2 for details).

2. Any thoughts on why HIF1 and HIF2 both regulate GPRC5A in cell lines, but only HIF1 does so in vivo?

We cannot rule out a role for HIF-2 in vivo, but a predominant role for HIF-1 may reflect the higher expression levels and stabilisation of the HIF-1a isoform in this specific context (i.e. on a background of Apc loss in the intestine). As noted by Newton et al. (PMID 20844082 and cited in the text), loss of Apc results in increased expression of Hif1a. Our data would indicate that this drives GPRC5A expression, as deletion of Hif1a on an Apc depleted background results in a marked reduction in Gprc5a mRNA (Figure 2D).

3. Higher magnification images of Figure 2C are needed to show sufficient detail.

We have now added additional/higher magnification images in Figure 2C (and EV2) as requested.

4. The data in Figure 2G-I showing a correlation between hypoxia, GPRC5A expression, and CRC patient survival are of somewhat limited value. The Kaplan-Meier curves may be particularly misleading, as GPRC5A expression in this context could simply be a surrogate for general HIF activation (see Kaelin, WG, (2017) Nature Rev Cancer 17: 425), as opposed to reflecting any functional role. This point should probably be acknowledged.

To accommodate the referee we have added the suggested reference and addressed the referee's point by adding the following sentence into the body text:

However, while these data show an in vivo association between GPRC5A, hypoxia gene signatures and patient outcomes, it is important to note this may be a reflection of GPRC5A's regulation by HIF activity/hypoxia in aggressive tumours, rather than necessarily indicating a functional role (Kaelin, 2017).

Referee #2 (Remarks for Author):

Greenough et al identified GPCR5A as a hypoxia inducible gene in proteomic analysis of colonic cancer cell line. They showed that GPCR5A is a direct and shared target of HIF1a/2a. The authors showed that GPCR5A overexpression protected cells from hypoxia-induced apoptosis (by Caspase WB). Bioinformatic analysis of colorectal cancer datasets revealed indicated that GPCR5A expression correlated with YAP target gene signatures. Departing from this point the authors conduct a series of experiments in which they show that hypoxia decreases YAP S397 and activates

YAP target genes, one of which is BCL2L1, in a GPCR5A-dependent way. Expression of the degron-resistant YAP S127A rescues caspase activation in GPCR5A knock down hypoxic cells.

The manuscript presents a very interesting observation linking hypoxic survival to GPCR5A through YAP activation. The experiments are technically superb and the conclusions are supported by the data. This is an important link between hypoxia and YAP function which is novel.

We were pleased that Referee #2 found our work to be very interesting and novel. We also pleased the referee thought our experiments were technically superb and that our data supported our conclusions.

There are specific aspects of this work that need to be clarified.

1) The paper does not address at all the mechanism by which GPCR5A alters YAP phosphorylation. Hypoxia was shown to activate YAP by degrading LATS2, in a SIADH2-dependnet way. Does GPCR5A signals through LATS1/2? Does it prevent LATS2 degradation? Is this done through Rho GTPases and if yes which one (RhoA/B/C). There is a need for biochemical experiments to address this mechanistic issue.

Although Referee #1 commented that the mechanistic details of how GPCR5A alters YAP phosphorylation are beyond the scope of our paper, to accommodate Referee #2 we have performed additional experiments to address the mechanism of how GPCR5A signals to YAP. We present new data in Figure 4D where we have examined the expression of phosphorylated (active) LATS1 (as well as total LATS1 and LATS2 levels) in response to hypoxia, with and without GPCR5A depletion. Consistent with YAP stabilisation by its dephosphorylation at Ser397 during hypoxia, we found that both activated (phosphorylated) LATS1 and total expression levels of LATS1/2 decreased during hypoxia. Importantly, these phenotypes were prevented by GPCR5A depletion (Figure 4D). This suggests that in hypoxia, GPCR5A depletion may stabilise and/or activate LATS1/2, leading to increased YAP phosphorylation.

To accommodate Referee #2's point regarding Rho GTPases, we now include data showing that overexpression of a constitutively active form of RhoA (G14V) reverses the effect of GPCR5A depletion on YAP phosphorylation (Figure 4E). Given that active RhoA (G14V) overrides the effect of GPCR5A depletion on YAP phosphorylation, our new data suggest that GPCR5A signals to YAP via RhoA. In support of this, we also show that expression of a dominant negative RhoA (T19N) does not lead to further increases in phosphorylated YAP in GPCR5A depleted cells (Figure EV4E). These data suggest that RhoA links GPCR5A to YAP in hypoxia.

2) The effect of GPCR5A on cell growth during hypoxia is inferred, based on the expression of cleaved PARP or activated caspase. There are NO actual cell growth data (crystal violet assays) with cells in which GPCR5A and/or YAP are manipulated in ways similar to the ones that produced the caspase changes. This is an important detail that will confirm the biochemical observation transates into active cell growth differences.

We agree with the referee that this was an important omission. To accommodate the referee, as requested we have performed several experiments that use crystal violet assays to measure cell growth/survival. These are: Figure 3C, showing cell growth/survival data in cells depleted of GPCR5A using three independent siRNA sequences in normoxia and hypoxia; Figure 4F, showing that YAP pathway inhibition (with YAP/TEAD inhibitor Verteporfin) preferentially reduces cell growth/survival in hypoxia; Figure 4H, showing that GPCR5A and YAP depletion affect cell growth/survival in hypoxia and that the GPCR5A depletion phenotype can be rescued by expression of an siRNA resistant GPCR5A cDNA. Note that we have also quantified the percentage of live, apoptotic and dead cells responding to GPCR5A manipulation by flow cytometry (Figure 3F).

3) The authors should test weather inactivation of YAP pathway in cells growing in hypoxia is sufficient to promote apoptosis.

We have included data that address the referee's comment and confirm that inactivation of the YAP pathway in cells growing hypoxia is sufficient to promote apoptosis. These data are shown in Figure 4F (inactivation of the YAP pathway with Verteporfin), Figure 4G and Figure 4I (YAP depletion increases markers of apoptosis cleaved caspase-3 and PARP). These data complement and support our existing data that show expression of constitutively active YAP (S127A) rescues hypoxic cells from apoptosis (Figure 4K).

4) Is non-hypoxic overexpression of GPCR5A sufficient to increase YAP S397 phosphorylation?

Having generated data from a cell line stably expressing a doxycycline-inducible GPCR5A cDNA construct, we are able to address this point by referring the referee to Figure 4I. Although we cannot rule this out, at least in our system we have not seen marked changes in YAP Ser397 phosphorylation upon expression of GPCR5A in normoxia; however, in hypoxia, doxycycline-induced GPCR5A^{siR} may enhance YAP Ser397 dephosphorylation (Figure 4I) and suggest that hypoxic conditions may be required for GPCR5A to signal via YAP. These data would be consistent with a GPCR5A-YAP signalling axis that promotes cell survival under conditions of hypoxia.

2nd Editorial Decision

05 July 2018

Thank you for the submission of your revised manuscript to EMBO Molecular Medicine. We have now received the enclosed report from the referee who was asked to re-assess it. As you will see the reviewer is now supportive and I am pleased to inform you that we will be able to accept your manuscript pending final editorial amendments.

***** Reviewer's comments *****

Referee #1 (Comments on Novelty/Model System for Author):

The revised manuscript is greatly strengthened by the addition of corroborating data, as well as new mechanistic insights, that strongly support and extend the authors' initial hypotheses and interpretations. This is a well-performed and well-controlled body of work that makes a novel connection between hypoxia, HIFs, an orphan GPCR, and the YAP signaling cascade, and which describes new mechanisms by which hypoxic responses regulate cell viability. It's a very solid and convincing story that adds an intriguing facet to our understanding of how tissue hypoxia regulates tumor progression.

Referee #1 (Remarks for Author):

The authors have done a commendable and very thorough job responding to the previous suggestions and critiques, and I believe the revised paper is certainly appropriate for publication in EMBO Molecular Medicine. I fully support its acceptance without further revision. Congrats on a really nice story.

Corresponding Author Name: Dr. Alexander Greenhough

Manuscript Number: EMM-2017-08699-V3